# Effectiveness of the e-NurSus Children Intervention in the Training of Nursing Students

**DOI:** 10.3390/ijerph16214288

**Published:** 2019-11-05

**Authors:** Cristina Álvarez-García, Carmen Álvarez-Nieto, Janet Kelsey, Rachel Carter, Sebastián Sanz-Martos, Isabel M. López-Medina

**Affiliations:** 1Department of Nursing, University of Jaén, 23071 Jaén, Spain; cagarcia@ujaen.es (C.Á.-G.); ssanz@ujaen.es (S.S.-M.); imlopez@ujaen.es (I.M.L.-M.); 2School of Nursing and Midwifery, University of Plymouth, Plymouth PL4 8AA, UK; j.a.kelsey@plymouth.ac.uk (J.K.); r.carter@plymouth.ac.uk (R.C.)

**Keywords:** child health, environmental health, nursing degree education, nursing students, paediatric nursing, questionnaire

## Abstract

The paediatric population is the most vulnerable to exposure to environmental risk factors. Institutions of higher education have to equip nursing students with the attitudes, knowledge, and skills to respond to this using blended learning. The aim of this study was to evaluate the effect of e-NurSus Children intervention on student nurses’ attitudes, knowledge, and skills. A quasi-experimental study of time series was designed using pre and post educational intervention evaluation in 2018. The participants were nursing students (*N* = 267) from Spain (*n* = 110) and the United Kingdom (*n* = 157). Three instruments were used: the Sustainability Attitudes in Nursing Survey, the Children’s Environmental Health Knowledge Questionnaire, and the Children’s Environmental Health Skills Questionnaire. The attitudes (15.81%), knowledge (39.02%), and skills (29.98%) of nursing students improved following the e-NurSus Children intervention. It is necessary to include topics on children’s environmental health in nurse education as students are aware of this issue but do not have the knowledge or skills required to manage problems or illness caused by the environment. The e-NurSus Children intervention is an effective tool to address this educational gap.

## 1. Introduction

Scientific evidence has shown that the environment poses a great risk to health. The climate is changing, including the intensity and frequency of extreme weather events, such as floods, heat, or cold, changes in rainfall causing either drought or torrential rains, rising sea levels, ocean acidification, worsening air quality, and longer pollen seasons [1,2,3,4,5]. Diseases caused by traditional forms of environmental pollution (e.g., coliforms in water or air pollution from solid fuels) are predominantly diarrhoea, pneumonia, and other infectious diseases. Modern environmental threats, by contrast, are linked mainly to non-communicable diseases, such as asthma, neurodevelopmental disorders, birth defects, obesity, diabetes, cardiovascular disease, mental health problems, and paediatric cancer [1]. The paediatric population, especially children under five years old, is the most vulnerable to exposure to environmental risk factors because of their immaturity and psychosocial dependence [6]. Children in rapidly industrializing countries are simultaneously confronted by both ancient and modern environmental threats to health [1].

Physical, chemical, and biological environmental hazards are responsible for 26% of all deaths in children under the age of five years, nearly 1.5 million deaths worldwide. This includes road accidents, ultraviolet and ionizing radiation, noise, electromagnetic fields, psychosocial risks, built environments, agricultural methods and man-made climate and ecosystem change, as well as pollution [1]. Specifically, air, water, soil, and toxic chemical pollution were responsible for 940,000 deaths in children [1]. The paediatric population, although it represents only 20–25% of the total number of people, includes 40–45% of all asthma cases [6] with the estimated overall worldwide prevalence of current asthma being 11.7% among children six to seven years old and 14.1% among those 13 to 14 years old [7].

The World Health Organization (WHO) estimates that approximately one-third of the disease burden in developing countries can be attributed to modifiable environmental factors, including indoor and outdoor air pollution, unsafe water, inadequate sanitation, and poor hygiene. The Lancet Countdown on Climate Change and the International Council of Nurses have recommended the integration of environmental health into nursing practice, education, research and policy/advocacy work [8]. Therefore, health professionals have an important role to play in understanding and communicating environmental-health-related issues to children and their families. It is recommended that nurses are highly educated in children’s environmental health in order to prevent and treat conditions that have been caused by environmental issues. However, most undergraduate and postgraduate nursing education programs do not address this issue, making it difficult to improve and influence the development of prevention strategies which target environmental illnesses or problems [4].

The environmental problems that society face make it necessary for higher education to equip its learners with the attitudes, knowledge and skills to pioneer innovative and creative responses in order to achieve wider economic, social and environmental well-being [8,9,10]. The attitudes, knowledge and skills needed to address climate change were defined by Álvarez-Nieto et al. [11].

One of the most important attitudes is to have vision and orientation towards the future, assuming high levels of responsibility, commitment to the environment and orientation towards change [11]. Nurses as advocates also have a responsibility to provide the public with evidence-based information about the health effects of chemicals and toxins and ways to reduce exposure. Nurses should be able to promote sustainable transport, energy savings, and the suitable placement of new schools [12]. The use of social media in health promotion has been found to reduce geographical barriers, however the effects on health outcomes are mixed [13]. Ultimately, management plans or changes in health policies should be promoted in order to achieve these objectives [11].

It is clear, therefore, that nurses need to acquire knowledge related to globalization and environmental impact. There is a necessity to explore how the health of individuals, communities, and populations are effected by social, political and environmental factors and to describe the interaction of the environment and child health at different levels [11]. This knowledge has to be developed together with ensuring nurses are confident in prevention and control strategies to develop interdisciplinary interventions [12]. In addition, there is a need for knowledge of the international and national legislation, all within the framework of health inequalities and the social determinants of health [11].

In terms of skills, nurses should have the communication skills to carry out environmental health counselling and education. This also implies having skills for dialogue and influence in other groups and sectors, as well as for awareness raising and citizen mobilization. It is recommended that they reflect on the consequences of environmental hazards and the possible interventions which should be developed from holistic, ethical and critical perspectives [11]. In addition, nurses should develop assessment skills to detect risks in children homes, schools, and communities using the environmental health history [12].

Nurses are agents of change who have the power to promote health and control the use of health resources [3,4]. Therefore, improving the level of attitudes, knowledge and skills on sustainability and the effects of climate change on the nursing profession will eliminate this educational gap and make possible a change in attitudes and modes of action in their daily work, which will result in the improvement of people’s health [10]. So, it is fundamental that nurses achieve sustainable literacy, including embedding key competencies in sustainability in the nursing curriculum [2,10,14,15].

E-learning is considered a good methodological strategy for education on children’s environmental health as it offers an alternative method of education [16,17]. It has recently become an increasingly used format for the education of health professionals [16,18,19,20] in order to move away from “instructor-centred, learner-passive” teaching to “student-centred, integrative learning” activities [21]. The meta-analysis by Voutilainen et al. [17] demonstrated that e-learning methods resulted in test scores that were 5.24 points higher than a conventional method on a 0–100 scale, clinical thinking was the only subject which was significantly better improved with conventional learning such as case-based learning (CBL) [16,17]. To overcome the difficulties presented by pure e-learning, blended learning has emerged and is defined as the combination of online material with students’ real-life experiences in face-to-face settings using CBL [20,21,22]. Studies [20] has showed that students enrolled on a blended model reported generally good experiences in linking online theory to practice. Advances in e-learning technology allow this to be organized in an online platform where the case study is uploaded and the issues discussed with the students in a face to face situation. [22].

For the purpose of this study, blended learning will also be referred to as the Blended On-Line and Digital (BOLD) [20]. In this learning methodology, students read online material prior to the class and it has been reported to enrich the learning [20,21,22]. Currently there is a lack of suitable e-products specially adapted for nursing education on environmental sustainability and health [23].

The authors of this study used the digital materials on environmental sustainability and nursing developed by NurSusTOOLKIT: A Teaching and Learning project—Resource for Sustainability in Nursing [24]. These materials are the only ones available that offer sustainability literacy and competency. They introduce the various concepts of health, nursing care, and sustainability, the relationships between health and the environment, healthy communities, and social and policy context.

A comparative study by Richardson et al. [25] looking at student nurses from four European countries (United Kingdom (UK), Spain, Switzerland, Germany), found that there were significant differences in attitudes toward sustainability, inclusion in the nursing curricula and implementation of sustainability in clinical practice. Although this international study assessed attitudes toward the environment, previous studies have not assessed knowledge and skills. Therefore, the aim of this study was to evaluate the effect of the e-NurSus Children intervention on student nurses’ attitudes, knowledge and skills. Based on previous theoretical studies [11,14] we hypothesized that the e-NurSus Children intervention delivered to nursing students would increase attitudes, knowledge, and skills to manage children’s environmental health problems.

## 2. Materials and Methods

### 2.1. Design

The study was designed as a quasi-experimental study of time series using pre and post educational intervention evaluation. It follows the STROBE (Strengthening the Reporting of Observational Studies in Epidemiology) and TIDieR (Template for Intervention Description and Replication) guidelines.

### 2.2. Sample and Setting

The target population was made up of nursing students from one Spanish university and one British university. All students who expressed their desire to participate in a subject on child health at a Spanish and British university were included. Students who filled out questionnaires incompletely or inadequately were excluded. The sample size was established to be a minimum of 17 pairs to achieve 80% power based in the one previous pre-post study to measure the impact of a sustainability-focused, scenario-based learning educational intervention on the attitudes and knowledge of student nurses [26].

### 2.3. Data Collection

Students participated in a taught session entitled e-NurSus Children. As it included interaction with online digital materials using e-learning and CBL, a BOLD learning methodology was used (Figure 1). Nursing lecturers (tutors) with previous experience in sustainability and nursing received standardized training before the sessions. The first meeting was undertaken at the Spanish university and all the tutors (both from United Kingdom and Spain) attended an experimental session to ensure that it would always be delivered in the same way. Students were introduced to a case study involving an asthmatic child. They worked in groups of four-five discussing issues arising from the assessment, case analysis and evidence collection and suggesting solutions or care formulation. The tutor guided students to exercise critical thinking about the problems and to identify solutions. Further information seeking and critique of proposed care regimens from each group were encouraged to promote learning. All of the materials used in the sessions were available, in Spanish and English, in the module J3_A1 Children’s Health and Environment on the NurSusTOOLKIT platform [27]. There was a maximum of 20 students in every session. The duration of each session was 90 min.

In Spain, the data collection was conducted in January–February 2018 and in the United Kingdom it was conducted in April–June and September–October 2018. In Spain, the session was a mandatory seminar for second-year students in a computer room, whilst in the United Kingdom, the session was a seminar for all child health nursing students, to which students brought their own laptops or mobile phones. The data were collected using self-administered online tools. They included the following:(1)Sustainability Attitudes in Nursing Survey (SANS_2). This survey was designed to evaluate nursing students’ attitudes toward sustainability and climate change. The items range from one to seven on a Likert scale. Reliability analysis revealed a Cronbach’s alpha of 0.82, and the five items loaded on a single factor explained 58% of the variance [26].(2)Children’s Environmental Health Knowledge Questionnaire (ChEHK-Q). This measures nursing students’ knowledge of children’s environmental health. It demonstrates good fit and a reliability of 0.98 for items and 0.70 for people based on the Rasch Model [28].(3)Children’s Environmental Health Skills Questionnaire (ChEHS-Q). This measures nursing students’ skills pertaining to children’s environmental health. It demonstrates good fit and reliability of 0.87 for items and 0.76 for people, based on Andrich’s rating scale model [28].

These three tools were used to collect data two weeks before and just after the educational sessions.

### 2.4. Data Analysis

First, we calculated the descriptive values of the scores on the questionnaires. Then, we compared questionnaire scores on two occasions using a test of differences of means to detect statistically significant differences between scores and items. Next, we calculated the improvement percentages of the test scores. Finally, we used a test of differences of means to compare the questionnaire scores at the two universities. The normal data distribution was tested with the Kolmogorov–Smirnov test. The level of significance was established at 0.05. Data was entered into IBM SPSS Statistics 25 (IBM, Armonk, NY, USA) for analysis.

### 2.5. Ethical Considerations

This study was approved by the Institutional Review Board of the University of Jaén (ABR.17/1) and the University of Plymouth (17-18-889). An information sheet was given to the participants. If happy to participate students completed and signed an informed consent form prior to undertaking the session. Students were not obliged to participate and were reassured that this would not affect their progress and success on their course of study. Confidentiality of personal data was guaranteed.

## 3. Results

The sample included 267 nursing students, 110 in Spain and 157 in the United Kingdom (Table 1). The reliability of the scales in all cases exceeded a value of 0.70, except for the ChEHK-Q post-test in Spain (Table 2).

### 3.1. Effectiveness of the e-NurSus Children Intervention

After testing with the Kolmogorov-Smirnov test, the only data that followed normal distribution were those referring to the knowledge pre-test in Spain so non-parametric tests were used. In all cases, the attitudes, knowledge and skills of nursing students improved after the intervention with a large effect size (Table 3). In general, the e-NurSus Children intervention improved environmental health knowledge the most (39.02%), followed by skills (29.98%), and finally attitudes (15.81%). In Spain, environmental health skills improved the most (42.33%), followed by knowledge (37.61%), and attitudes (22.96%). In the United Kingdom, environmental health knowledge (40.38%) improved the most, followed by skills (21.70%), and attitudes (11.06%). Table 4 shows the differences in the Mann–Whitney *U* test between Spain and the United Kingdom. The attitudes and skills were higher in British students before the intervention but after the intervention they were higher in Spanish students. The knowledge was higher among Spanish students before and after the intervention.

### 3.2. Attitudes, Knowledge, and Skills for Addressing Climate Change

In Spain, all the items on SANS_2 (attitudes) showed an improvement in their mean value of attitude being statistically significant (Appendix A). The item that improved the most was item two (*inclusion of climate change in the nursing curriculum*) with an increase of 35.99%, (4.64 in the pre-test to 6.31 in the post-test). On ChEHK-Q (knowledge) most of the items improved with statistical significant differences. The item that showed the most improvement was item 22 (*food colourings and nervous system problems*) with an increase of 140.54% (from 37 correct answers on the pre-test to 87 in the post-test). Four items on this test obtained fewer correct answers on the post-test than on the pre-test; item five (*effects of nitrogen oxide from fossil fuels*) worsened by 42.86%, from 14 correct answers on the pre-test to 8 on the post-test. On the ChEHS-Q (skills), all items improved their mean being statistically significant; specifically, item five (*health education on contaminants in food*) improved its mean by 56.88% (from 2.76 on the pre-test to 4.33 on the post-test).

In the United Kingdom, all of the items on SANS_2 (attitudes) improved with statistical significance, except for item three, which improved without statistically significant differences (Appendix A). As in Spain, the item which improved the most was item two (inclusion of climate change in the nursing curriculum) with an increase percentage of 24.05%, from 4.74 in the pre-test to 5.88 in the post-test. On the ChEHK-Q (knowledge), most of the items improved with statistical significance; only two improved without statistical significance and five worsened. The item which improved the most was item 18 (*effects of water flowering*) with an increase of 135.29% (from 34 correct answers on the pre-test to 97 on the post-test). The item which worsened the most was item five (*effects of nitrogen oxide from fossil fuels*) with a decrease of 25.93% (from 27 correct answers on the pre-test to 20 on the post-test). On the ChEHS-Q (skills), all items improved their mean being statistically significantly different; specifically, item three (*identify environmental risks associated with childhood neoplasms*) was the one which improved the most by 60.40% (from 2.02 on the pre-test to 3.24 on the post-test).

The data showed that in Spain 97 (88.18%) students had good attitudes (excellent, very good and good) before the intervention but after the intervention the number increased to include all of them; 110 (100%). The data also showed that 99 students (90%) had good knowledge after the intervention, while only 43 participants (39.09%) had good knowledge before the intervention. Sixty-one participants (55.45%) had good skills after the intervention, while 79 (71.82%) had good skills before the intervention. In this case, although the overall average increased significantly, the number of participants with good skills decreased after the intervention. However, no participants with poor skills remained so the majority of students have good or insufficient skills. In the United Kingdom the data show that 151 students (96.18%) had good attitudes after the intervention, a number very similar to the number before the intervention (153 students, 97.45%). After the study, there were a higher number of participants with excellent attitudes and only one participant with poor attitudes after the intervention. The number of participants with good knowledge (101, 64.33%) was considerably higher after the intervention whereas before the study only 30 participants (19.11%) had good knowledge. As in the case of attitudes, the number of participants with good skills after the intervention was similar to before the intervention, 1 (0.65%) to 2 (1.27%) but the number of participants with poor skills decreased from 6 to 4. These data are detailed in Table 5.

## 4. Discussion

The aim of this study was to evaluate the effect of e-NurSus Children intervention on student nurses’ attitudes, knowledge and skills. The attitudes, knowledge and skills of nursing students improved following the e-NurSus Children intervention in both countries. The attitudes of nursing students towards the environment were excellent but the knowledge and skills to deal with environmental problems in childhood were shown to be only good. Nevertheless, the most improved areas in both countries were knowledge and skills.

The smaller percentage of improvement in attitudes may be because attitudes were already positive before the intervention. This is important to acknowledge because favourable environmental attitudes between nurses are seen as the first step in enhancing nurses’ and nursing students’ awareness about the risks which the environment could pose to health and how to prevent and treat conditions caused by environmental factors in applying sustainable care [26,29]. The British university students showed less improvement in attitudes, possibly because they had better attitudes before. However, the intervention was highly effective in Spain, as demonstrated by the percentage of positive attitudes was higher than in the United Kingdom. In the case of knowledge and skills, it was also found that Spanish students improved more. One explanation may be that the intervention was a compulsory seminar for Spanish students and children’s environmental health was included in the nursing curriculum and therefore it may be more effective than for those taking this intervention as a voluntary or optional topic. This view was supported by several authors in previous theoretical studies [11,14].

An important question raised was the fact that some items in the knowledge questionnaire received lower scores after the intervention. In both countries the item which reduced the most referred to *the effects on health of nitrogen oxide from fossil fuels*. Other items which reduced among Spanish students referred to *water containing nitrates, pesticides, and mercury*; for British students, apart from these topics, they displayed lower scores in the questions about *solar ultraviolet radiation* and *air pollutants*. However, topics where the students improved included *cancers and nervous system problems*. In the case of attitudes and skills, all of the items scored similarly demonstrating that they also improved similarly after the intervention. These results should be acknowledged in order to improve future interventions focusing on the topics that had the most reduced scores.

These results demonstrated an improvement in competencies, attitudes, knowledge and skills in children environmental health issues. The findings are similar to those found by Richardson et al. [8,10,30] and Grose et al. [31] in skills sessions with nursing students in the United Kingdom but they differ from those found by Richardson et al. [32] which revealed no statistical differences, however, this could possibly be explained by their small sample size. In Richardson et al. [30,32], SANS and SANS_2 were used respectively to assess the skills gained in a session related to sustainability three months after the intervention, so comparisons can be made. It is also possible to compare the result of Richardson et al. [8] with our results since SANS_2 was used to measure the attitudes of student nurses and midwives towards climate change and sustainability and the inclusion of these topics in nurse education. The results from this study showed higher scores on all the attitudes items, both before and after the interventions than in Richardson et al. [32], however, in Richardson et al. [26] the item referring to *sustainability as an important issue for nursing* was higher after the intervention than in this study. Otherwise, the results of this study were similar as British students identified *sustainability as the most important issue for nursing* and Spanish students assessed this item as the second most important issue. In the work of Richardson et al. [8] the students before the sessions had lower attitudes than our students but it is understandable because in this study the student nurses were in the first week of the nursing degree. However, after the sessions the second year students achieved better results than the British students in all the items except the item referring to applying sustainability principles at home. The third year students achieved better results than the British students in only two items, referring to *the importance of sustainability for nursing* and *the inclusion of it in the nursing curriculum*. These enhanced results can be explained as the students received several sessions as opposed to just one in this study and the improvement was more significant. However, it should be recognised that a single session managed to improve attitudes on some aspects more than several sessions during the years of education in nursing. This demonstrates the strength of the intervention as seen in this study.

Although all the results of previous studies in this field show an improvement in knowledge after interventions related to sustainability, they measure knowledge of sustainable care rather than children’s environmental health. Richardson et al. [10] and Grose et al. [31] assessed the knowledge of British nursing students after intervention with three questions referring to *knowledge of service delivery resources, management of waste in healthcare* and *awareness of peak oil*. In Richardson et al. [26] the knowledge acquired by the British students in a similar session was assessed with three questions referring to *plastic provenance, cost of clinical waste,* and *cost of domestic waste*. In Richardson et al. [32], knowledge of sustainability gained after the intervention was also assessed qualitatively. This study utilised more specific questionnaires which made it possible to detect deficit areas in environmental matters.

Robust results were achieved, underpinned by data from two universities in different countries with a large sample size. Previous studies assessed only attitudes towards the environment [26] and they omitted the important of a high level of knowledge and skills to manage environmental problems, besides the fact that most of the previous studies were focused in only one country [8,26,32] so it was difficult to translate the results into an international context. Despite the strengths of this study, it has some limitations. One is that social desirability may play a role in shaping participants’ answers when they were answering the attitudes and skills questionnaires, although this was limited because students were assured that the questionnaires would be anonymous and confidential. The second limitation is that the sample was derived in a non-random way, which might lead to selection bias. In order to minimize this bias, the values in the pre-test were compared between groups and the improvement percentages calculated in order to make a more objective comparison. In addition, the data collection process was not carried out in the same conditions with respect to the students’ access to computers which could have led to some information bias. However, the intervention protocol was carried out in the same manner, which allows conclusions to be drawn as to whether it was preferable to provide students with the technological devices that are required for the intervention or to use those they are familiar with.

This study was an initial step in enhancing nurse education in relation to sustainable development of environmental topics in futures nursing courses. It was just one example in one children’s nursing course and if developed further and wider across nursing curricula it has the potential to improve the knowledge and skills of future nurses in this hugely important area. Future projects should try to develop e-NurSus Children intervention within nursing curricula to improve student nurses’ attitudes, knowledge and skills in children’s environmental health in different universities internationally.

The Children’s Environmental Health project [33] is evaluating the attitudes, knowledge, and skills in children’s environmental health of nursing students from universities in different countries. Its next phase is to undertake educational interventions focused on the deficit areas. Future research, possibly in other universities, where the topic is not already included in the curriculum could include a control group and this could further explore and investigate the extent of the impact from education for sustainable development interventions in this specific area.

## 5. Conclusions

Nursing students are aware of the necessity to expand and develop their knowledge to include topics on children’s environmental health as at present do not have enough knowledge or the skill needed to manage problems or illness caused by the environment to a proficient level. The e-NurSus Children intervention is a meaningful way to improve attitudes, knowledge, and skills regarding children’s environmental health among the nurse student population. This interactive, blended-learning, educational intervention can easily be added into most nursing curricula to enhance learning in this vital area.

## Figures and Tables

**Figure 1 ijerph-16-04288-f001:**
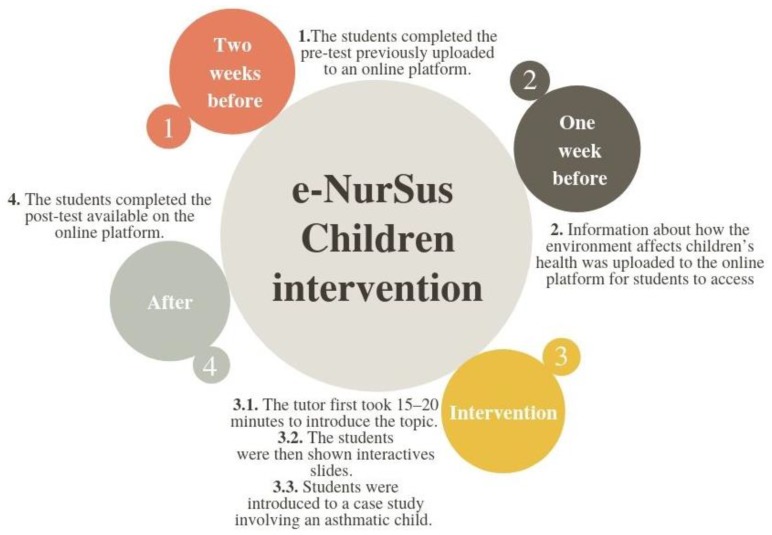
Protocol for the e-NurSus Children intervention.

**Table 1 ijerph-16-04288-t001:** Demographic characteristics of the samples.

Variable	Spain ^1^		United Kingdom ^2^	
Characteristic	*N*	%	*N*	%
Age (*M*/*SD*)	21.28	4.59	23.27	5.78
Gender				
Male	22	20.00	4	2.50
Female	88	80.00	153	97.50
Year of course				
First	0	0	43	27.40
Second	106	96.40	63	40.10
Third	4	3.60	51	32.50
Fourth	0	0		

^1^*n* = 110. ^2^
*n* = 157.

**Table 2 ijerph-16-04288-t002:** Reliability values measured with Cronbach’s alpha coefficient for the pre-test and post-test scales in Spain and United Kingdom.

Variable	Pre-Test		Post-Test	
Scale	Spain	United Kingdom	Spain	United Kingdom
SANS_2	0.836	0.857	0.756	0.894
ChEHK-Q	0.714	0.813	0.652	0.813
ChEHS-Q	0.863	0.878	0.915	0.924

SANS_2: Sustainability Attitudes in Nursing Survey; ChEHK-Q: Children’s Environmental Health Knowledge Questionnaire; ChEHS-Q: Children’s Environmental Health Skills Questionnaire.

**Table 3 ijerph-16-04288-t003:** Means of attitudes, knowledge, and skills pre-test and post-test.

Variable		*M*		Wilcoxon Matched-Pairs Signed-Rank Test		
Scale	*n*	Pre-Test	Post-Test	*z*	*p*-Value	*d*
Total						
SANS_2 (attitudes)	267	26.50 ± 5.68	30.69 ± 4.59	−10.12	<0.001	0.81
ChEHK-Q (knowledge)	267	12.66 ± 4.46	17.60 ± 4.27	−12.28	<0.001	1.13
ChEHS-Q (skills)	263	36.86 ± 8.51	47.91 ± 8.02	−13.03	<0.001	1.34
Spain						
SANS_2 (attitudes)	110	25.70 ± 5.78	31.60 ± 3.27	−8.05	<0.001	1.26
ChEHK-Q (knowledge)	110	14.17 ± 3.81	19.50 ± 3.29	−8.47	<0.001	1.50
ChEHS-Q (skills)	109	35.81 ± 9.54	50.97 ± 7.01	−8.89	<0.001	1.81
United Kingdom						
SANS_2 (attitudes)	157	27.06 ± 5.55	30.05 ± 5.23	−6.23	<0.001	0.55
ChEHK-Q (knowledge)	157	11.59 ± 4.59	16.27 ± 4.38	−8.89	<0.001	1.04
ChEHS-Q (skills)	154	37.60 ± 7.63	45.76 ± 8.01	−9.35	<0.001	1.04

**Table 4 ijerph-16-04288-t004:** Means of attitudes, knowledge, and skills in Spain and the United Kingdom.

Variable		*M*		Mann-Whitney *U* Test		
Scale	*n*	Spain	United Kingdom	*z*	*p*-Value	*d*
SANS_2 (attitudes)						
Pre-test	267	25.70 ± 5.78	27.06 ± 5.55	−2.138	0.032	0.24
Post-test	267	31.60 ± 3.27	30.05 ± 5.23	−2.106	0.035	0.36
ChEHK-Q (knowledge)						
Pre-test	267	14.17 ± 3.81	11.59 ± 4.59	−4.415	<0.001	0.61
Post-test	267	19.50 ± 3.29	16.27 ± 4.38	−6.602	<0.001	0.83
ChEHS-Q (skills)						
Pre-test	263	35.81 ± 9.54	37.60 ± 7.63	−2.157	0.031	0.21
Post-test	263	50.97 ± 7.01	45.76 ± 8.01	−1.040	0.298	-

**Table 5 ijerph-16-04288-t005:** Number of participants with excellent, very good, good, insufficient, or poor attitudes, knowledge, and skills in pre-test and post-test.

Measure	*n*	Excellent	Very Good	Good	Insufficient	Poor
University of Jaén						
Pre-test						
Attitudes	110	23 (20.91%)	37 (33.64%)	37 (33.64%)	13 (11.81%)	0 (0%)
Knowledge	110	0 (0%)	6 (5.45%)	37 (33.64%)	47 (42.73%)	20 (18.18%)
Skills	109	3 (2.75%)	50 (45.87%)	26 (23.86%)	27 (24.77%)	3 (2.75%)
Post-test						
Attitudes	110	64 (58.18%)	41 (37.27%)	5 (4.55%)	0 (0%)	0 (0%)
Knowledge	110	3 (2.73%)	50 (45.45%)	46 (41.82%)	8 (7.27%)	3 (2.73%)
Skills	109	1 (0.92%)	0 (0%)	59 (54.13%)	49 (44.95%)	0 (0%)
University of Plymouth						
Pre-test						
Attitudes	157	41 (26.12%)	69 (43.95%)	42 (26.75%)	2 (1.27%)	3 (1.91%)
Knowledge	157	0 (0%)	2 (1.27%)	28 (17.83%)	70 (44.59%)	57 (36.31%)
Skills	154	0 (0%)	0 (0%)	1 (0.65%)	147 (95.45%)	6 (3.90%)
Post-test						
Attitudes	157	71 (45.22%)	66 (42.04%)	14 (8.92%)	5 (3.18%)	1 (0.64%)
Knowledge	157	3 (1.91%)	22 (14.01%)	76 (48.41%)	39 (24.84%)	17 (10.83%)
Skills	154	0 (0%)	0 (0%)	2 (1.30%)	148 (96.10%)	4 (2.60%)

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
