# Peer review of "Effectiveness of the e-NurSus Children Intervention in the Training of Nursing Students"

_ijerph, 2019, doi:10.3390/ijerph16214288_

Round 1
Reviewer 1 Report
Recently, the e-learning education for children care for nurse students is emphasized as one of the important education strategy. In that sense, this paper is also considered a well-written paper. However, the paper is provided to readers and sharing the paper's results is also important for the reproducibility of the paper's results. However, the structure and contents of the experiments are not described in detail in this paper. And it is not written what kind of theoretical background these experimental studies include. Therefore, it is better to present the entire contents of the experimental program in the the research as a figure, and describe the theoretical background of the necessity part of the research.
Reviewer 2 Report
Thank you for opportunity for reviewing this paper. The manuscript is generally well written, but it is unclear what this review adds to what is already known and have been published earlier. No clear research question seems to be formulated, and the conclusions are unclear. I have read it with interest, but regretfully have decided to reject it as unsuitable for this particular journal and these reason will be discussed below.
- Title: The title is correct as it reflects correctly the objective of the work.
- Summary: Correct
- Introduction: The introduction section is too short, weak and did not provide a clear rationale for carrying out the study (for example, why is your research question important? What gap in the literature is the study addressing?
- I suggest in this section should be improved, with more details about prevalence, impact related with this question.
Also, please describe the hypothesis and objectives in this section.
- Material and Methods: This section is poor, needs to present a better rationale for the study and the methodology employed. Also, neither appear information related with inclusion and exclusion criteria, dates, protocol. The study design is a experimental research of ramdom sampling method, where the study was conducted in the university center? This research adhere to reporting CONSORT guidelines?
Likewise more detail about information calculate sample size and data should be provided. Also, please need include the data and record code and all information related with registered of this research at ClinicalTrials.
Results: The results in basis of the used method are not informative. I dont believe this study adds a great deal of novel and new information.
Discussion: I am struggling to make sense of some of this, I am afraid it needs extensive revision. What are the clinical and non clinical implications of your study? How this will inform future larger studies?
Conclusion: These conclusions is too long and need to be softened, modified in order to reflect only the study findings.
Reviewer 3 Report
There is a need to integrate climate change into nursing curricula worldwide. Relevant study. The abstract a clear aim but the introduction section did not....usually this section will conclude with a clear purpose/aim statement and research questions if applicable. Suggest authors use more tentative language throughout manuscript. For example, lines 44-45 suggest removing "must" ..."Nurses must..." and instead use "It is recommended...."
Method section: authors identify the instruments of measure and it appears report the Chronbach's alpha for this study. These are results and should be reported in the Result section of this research manuscript. Authors are encouraged to report the Crohnbach's alpha on these instruments in previous studies in the Method section.
IRB approval- please detail if students were informed that by not participating in this study it would not impact their course grade.
Discussion section: Lines 249 Clarify "The main....(what)"
Limitations: Was there a standardized training for tutors? How was this done...what was their specific role...this could be a limitation if not standardized.
Round 2
Reviewer 2 Report
I would like to thank the authors for their work, however I did not feel the authors made any significant improvements with regards to the main issues I raised in the first review. In its present state the paper provides no clear evidence that the authors propose a experiment must have been conducted rigorously, due to the justification and significance for the current piece of research need it be clearly evidenced. In addition, the choice of participants, sample size, study design and choice of outcome measures need to be given rationale. Discussion and conclusion need to focus on the findings and its significance.
Also, this research must should register in clinical trials due that is a clinical study involves research using human volunteers (also called participants) that is intended to add to nursing knowledge. There are two main types of clinical studies: clinical trials (also called interventional studies) and observational studies. ClinicalTrials.gov includes both interventional and observational studies with a record number.
I regret that the disposition is not favorable, but would like to thank you for your support.
We wish you all the best.
